# Impact of stress and coping strategies on quality of life in hematological malignancies: A cross-sectional study

Yue Qi[1], Kangsheng Zhu[2], Wei Liu[1], Xiaolei Xu [3]*

1 Department of Hematology, the Fourth Hospital of Hebei Medical University, Shijiazhuang, China,
2 Department of Anesthesiology, the Fourth Hospital of Hebei Medical University, Shijiazhuang, China,
3 Department of Clinical Nursing Teaching and Research Section, School of Nursing, Hebei Medical University, Shijiazhuang, China

* 18801449@hebmu.edu.cn

## Abstract

### Background and aims

While perceived stress and coping strategies have been established as significant determinants of quality of life (QoL) in patients with solid malignancies, their impact on hematological malignancy population have not been fully elucidated. This study aimed to examine how perceived stress and medical coping strategies interact with sociodemographic factors to influence QoL in patients with hematologic malignancies.

### Methods

The study, involving 185 hematologic cancer patients in China, was conducted between August 2024 and December 2024. Self-reported scales were used to assess QoL, perceived stress and medical coping strategies. Data analysis included univariate analyses, Pearson correlation, and multivariate regression analyses using SPSS V26.0.

### Results

178 patients completed the survey with a response rate of 96.2%. The overall QoL score was 55.99±24.6 indicating a moderate-to-low level. The functional domains averaged 64.81±21.83, while the symptomatic domains averaged 24.95±19.38. The overall QoL was associated with age, sex, marital status, place of residence, type of insurance ($p<0.05$). Stress perception (r=−0.389, −0.591), crisis perception (r=−0.489, −0.638), and yielding coping (r=−0.479, −0.547) exhibited significant negative correlations with overall QoL and functional scores, respectively ($p<0.001$). Stress perception (r=0.435), crisis perception (r=0.535), avoidance (r=0.280), and yielding (r=0.472) were positively correlated with symptom scores ($p<0.001$).

**Data availability statement:** All relevant data are within the manuscript and its Supporting Information files.

**Funding:** Medical Science Research Project of hebei for 2024 (Project number 20241949).

**Competing interests:** The authors have declared that no competing interests exist.

Multiple linear regression analysis identified sex, crisis perception, and yielding as key factors affecting overall QoL (explaining 36.5% of the variance) and functional status (explaining 45.6% of the variance), while residence, crisis perception, avoidance, and yielding significantly influenced the symptomatic fields (34.2% variance) in patients with hematological malignancies ($p < 0.05$).

## Conclusions

This study revealed that the QoL of patients with hematological malignancies is generally suboptimal and influenced by sex, crisis perception, and stress-coping strategies. These findings underscore the need for integrated psychosocial interventions targeting stress management and adaptive coping strategies in clinical care. This study contributes to the growing body of evidence on psychosocial determinants of QoL in oncology and can inform tailored supportive care programs for this population.

## 1 Introduction

Hematological malignancies comprise a variety of cancers derived from the blood, bone marrow and lymphatic system. Over the past 30 years, the incidence and deaths from haematologic malignancies have increased significantly, even ranking as the fourth most frequently diagnosed cancer [1–3]. According to recent global data, haematologic malignancies represented 6.6% of all cancer cases and 7.2% of cancer-related deaths, with a lifetime risk of 1.67% for incidence and 0.98% for mortality, respectively [4,5]. The prognosis for patients with hematologic tumors varies depending on factors such as tumor type, stage, patient age, and overall physical condition. Recent advancements in molecular biology and immunology have expanded treatment options, offering renewed hope for survival [6,7]. However, despite improved clinical outcomes, patients often experience persistent symptoms and functional impairments post-treatment, including fatigue, pain, nausea, and reduced quality of life (QoL) [8,9]. These challenges exacerbate economic burden and contribute to psychological distress, such as anxiety and depression [2,10]. Moreover, the inherent unpredictability of treatment outcomes and prognostic uncertainty further exacerbates patients' psychological vulnerability through persistent mortality threats [11].

Cancer patients' QoL is substantially affected by perceived stress and coping strategies, even sociodemographic factors [12,13]. Studies indicate that higher perceived stress is associated with lower QoL [14, 15]. Coping style moderates this relationship, with positive strategies attenuating and negative strategies amplifying stress-induced QoL impairment [16]. Individuals facing stress adopt various coping strategies, and some studies have suggested that effective coping strategies can alleviate patient stress and improve their QoL [17]. Growing evidence highlights the profound impact of perceived stress on QoL and established fundamental links between stress and QoL, yet most evidence comes from solid tumor studies [18–20]. Hematologic cancers, with their characteristic psychological and physical challenges, have received

comparatively little attention, particularly regarding how distinct perceived stress and coping approaches combine to shape patients' QoL. Therefore, understanding patients' perceptions of disease-related stress and their coping strategies from their perspective helps to uncover their genuine psychological needs and identify effective interventions to enhance their self-management abilities, ultimately improving their QoL.

Based on this, the present study focuses on patients with malignant hematologic diseases to explore how do perceived stress and medical coping strategies interact with sociodemographic factors, aiming to provide a theoretical basis for improving the QoL of hematologic cancer patients. Our findings will inform targeted psychosocial interventions to mitigate stress and promote adaptive coping, ultimately improving patient-centered outcomes.

## 2 Materials and methods

### 2.1 Study design and ethical considerations

This study strictly followed the STROBE guidelines (see S1 Table). From August 2024 to December 2024, a cross-sectional study was undertaken from the department of Hematology of the fourth hospital of hebei medical university. Although a single-center sample may limit the wide applicability of the results, this hospital is one of the largest hemato-logical oncology centers in northern China. It is a member of the Lymphoma Collaboration Group of Northern China and the Lymphoma Working Group of China, which has a substantial pool of hematology patients and follows standardized treatment protocols, while strictly controlling for confounding variables to enhance the reliability of causal inferences. This study received approval from the Institutional Review Board (IRB) of the Research Ethics Committee (IRB no: 2024KS107, Hebei Medical University Fourth Hospital Medical Ethics Committee). We did not include minors, and all participants provided written informed consent. Participants signed a written informed consent form before enrollment, which covered the research purpose, expected time consumption, privacy protection measures (data anonymization), and the right to voluntarily withdraw.

### 2.2 Participants

A total of 185 questionnaires were distributed to the Hematology Department of the Fourth Hospital of Hebei Medical University. The inclusion criteria were as follows: (1) patients with hematological malignancies confirmed by clinical and pathological diagnoses; (2) patients aged 18 years or older; (3) patients receiving standard treatment within the hospital; (4) Patients who were aware of their diagnosis and treatment; (5) patients who were able to comprehend the purpose and significance of the study and were willing to voluntarily participate; and (6) patients with clear consciousness, normal comprehension, communication, and expression abilities. The exclusion criteria were as follows: (1) patients who were unable to cooperate in completing the study due to illness or personal reasons; (2) patients with unstable conditions or those who experienced rapid deterioration over a short period. Additionally, exclusions were made for (1) patients with incomplete questionnaires (2) patients who exhibited logical errors while completing the questionnaire.

### 2.3 Questionnaire survey

Each participant was fully briefed on the aims and procedures of the study, which included the completion of questionnaires detailing general patient information and the European Organization for Research and Treatment of Cancer Quality of Life Questionnaire (EORTC QLQ-C30), the Perceived Stress Scale (PSS-10), and the medical coping modes questionnaire (MCMQ). Upon completion, the questionnaires were promptly collected and reviewed. Any gaps, missing items, or logical inconsistencies were addressed immediately, with patients requested to make the necessary corrections according to their discretion to ensure the completeness and validity of the data. After recovery, the questionnaires were scrutinized by two researchers and assistants, with any invalid questionnaires excluded. The data entry process was then conducted, followed by a secondary review of the data by the two researchers to ensure accuracy.

### 2.4 Measurements

**2.4.1 Demographic questionnaire.** The general information questionnaire was developed through consultation and reference to relevant research literature. The main content of the survey included general patient information such as sex, age, education level, marital status, residence, economic income, and type of medical insurance.

**2.4.2 EORTC-QLQ-C30.** The EORTC-QLQ-C30 is primarily used to assess the QoL of cancer patients [21,22]. It encompasses four dimensions, a functional scale, a symptom scale, single items, and general health status, comprising a total of 30 items. Items 1–28 were scored on a 4-point Likert scale ranging from "none" to "a lot," whereas items 29 and 30 utilized a 7-point Likert scale ranging from "very bad" to "very good." Functional scale scores were linearly transformed from 0 to 100 according to the scale's operational guidelines, with higher scores indicating better QoL. Higher scores in the functional domain and general health status reflect better QoL, whereas lower scores in the symptom domain suggest better QoL overall. The Chinese version of EORTC-QLQ-C30 was used to evaluate the quality of life of patients with hematological malignancies. This table has good reliability, validity and responsiveness, and is suitable for the determination of the quality of life of cancer patients in China [23]. In this study, the EORTC QLQ-C30 demonstrated good internal consistency, with an overall Cronbach's alpha of 0.73. All subscales exhibited satisfactory reliability ($\alpha > 0.7$), particularly the functional fields ($\alpha = 0.93$) and the symptomatic fields ($\alpha = 0.81$).

**2.4.3 PSS-10.** The PSS-10, developed by Cohen, is designed to evaluate the degree of perceived stress and the ability to cope with it [24]. The scale consists of 10 items divided into two dimensions: crisis perception and coping ability. Each item is scored from 0 to 4, indicating "never" to "always." Notably, the scores for items 4, 5, 7, and 8 are reversed, meaning that the higher the total score is, the greater the stress perceived by the individual. The scale of the (PSS-10) score ranges from 0 to 40, and stress levels can be classified into three grades based on the score: 0−13: Low stress level, indicating a relatively low stress level perceived by the individual in the past month. 14−26: Moderate stress level, indicating a moderate level of stress perceived by the individual in the past month. 27−40: High stress level, indicating a higher stress level perceived by the individual in the past month. The reliability and validity of the Chinese version of PSS-10 have been evaluated. Its Cronbach's α coefficient was 0.91 and the test-retest correlation coefficient was 0.69 [25]. In this study, the Cronbach's α values was 0.89.

**2.4.4 MCMQ.** The MCMQ was used to evaluate the coping styles of patients with this disease [26]. It contains 20 items across three subscales: yielding (7 items), confrontation (8 items), and avoidance (5 items). Each item is scored from 1 to 4, with higher scores indicating a greater inclination for the patient to adopt certain coping strategies in response to their illness. In this study, the Cronbach's α values was 0.85.

### 2.5 Sample size

According to the sample size estimation method, the sample size in multiple linear regression should be taken from 10 to 20 times the number of variables [27]. This study has a total of 10 dimensions, and the calculated sample size is approximately 100–200 cases. Given the possible potential loss (10–15%) of invalid samples or data, a total of 185 questionnaires were distributed in this study.

### 2.6 Statistical analysis

All analyses were performed using IBM SPSS Version 26.0 (SPSS Statistics V26.0, IBM Corporation, Somers, New York). The dataset used for analysis is available in Supporting Information as S1 Data. Continuous variables were reported as mean ± standard deviation (SD). Categorical variables were presented as number and percentage. Univariate analyses exploring associations between study variables and QoL were performed using Non-parametric test methods (Kruskal-Wallis test or Mann-Whitney U test). Pearson's correlation analysis was used to analyse the correlations among QoL, awareness of stress and medical coping strategies. Multiple linear regression analysis was used to identify the factors

associated with the dependent variables. The total score of overall QoL, functional fields and symptomatic fields were used as the dependent variables, and the statistically significant variables in the univariate and correlation analyses were used as the independent variables in the multiple linear regression analysis. Post hoc pairwise comparisons were conducted using Bonferroni adjustment for multiple comparisons. The value of R squared was calculated and the statistical significance level for all the tests was set at $p < 0.05$, two-tailed.

## 3 Results

### 3.1 Demographics and characteristics of the participants

A total of 185 questionnaires were sent out in this study. The questionnaires that were incomplete and did not meet the inclusion criteria (not blood disease) were excluded. A total of 178 patients completed the survey, yielding a response rate of 96.2%. The mean age of these patients was 54.27 ± 15.01 years. Among these patients, 40.4% were females. Various education levels, marital statuses, places of residence, types of insurance and cancer sites were represented. As shown in Table 1.

**Table 1. Demographic and clinical characteristics of the participants.**

| Social information | |
|---|---|
| Age, mean (SD) | 54.27 ± 15.01 |
| Sex, n (%) | |
| Male | 106 (59.6) |
| Female | 72 (40.4%) |
| Education level,n (%) | |
| Primary school | 28 (15.7%) |
| Junior high school | 74 (41.6%) |
| Senior high school | 41 (23.0%) |
| Above high school | 35 (19.7%) |
| Marital status,n (%) | |
| Married | 151 (84.8%) |
| Unmarried | 11 (6.2%) |
| Divorced | 3 (1.7%) |
| Widowed | 13 (7.3%) |
| Place of Residence,n (%) | |
| City | 47 (26.4%) |
| County | 27 (15.2%) |
| Towns and villages | 10 (5.6%) |
| Village | 94 (52.8%) |
| Type of insurance,n (%) | |
| Urban medical insurance | 51 (28.7%) |
| New rural cooperative medical system (NCMS) | 108 (60.7%) |
| Other | 19 (10.7%) |
| Clinical information at study entry,n (%) | |
| lymphoma | 145 (81.5%) |
| MULTIPLE myeloma | 24 (13.5%) |
| Leukaemia | 6 (3.4%) |
| Other | 3 (1.7%) |

## 3.2 EORTC QLQ-C30, PSS-10 and MCMQ scores of the study participants

The results of the EORTC QLQ-C30 are shown in Table 2. The EORTC QLQ-C30 is primarily divided into three major categories: functional scales, symptom scales, and the quality of life scale. For functional scales and quality of life, higher scores indicate better functioning or quality of life. For symptom scales, higher scores indicate more severe symptoms. Interpretation of quality of life scores: ≥ 70 points: Good quality of life. 40−69 points: Moderate level. < 40 points: Poor quality of life. The overall quality of life score was 55.99 ± 24.64 indicating a moderate-to-low level. In the functional field, the social function score (48.60 ± 31.09) was the lowest. In the symptomatic group, the fatigue score (40.01 ± 25.34) was the highest, and the nausea and vomiting score (13.39 ± 20.05) was the lowest. In the 6 single-item fields, the score for financial difficulty (61.24 ± 35.46) was the highest, and the score for diarrhea (9.18 ± 17.98) was the lowest. The PSS-10 total score was 19.63 ± 4.35, crisis perception and coping ability score was 11.06 ± 3.90 and 8.57 ± 2.20. The confrontation, avoidance and yielding score was 18.87 ± 4.02, 16.87 ± 2.69 and 9.56 ± 3.27, respectively. As shown in Table 2.

## 3.3 Univariate analysis of quality of life among patients with hematological cancer

Age, sex, marital status, residence, and medical insurance type affected the overall life quality score ($p < 0.05$).Sex and medical insurance type affected the functional and symptom field scores ($p < 0.05$). As shown in Table 3.

## 3.4 Correlation analysis of quality of life, awareness of stress and medical coping strategies in patients with hematological diseases

The stress perception, crisis perception and yielding exhibited a negative correlation with quality of life and function scores ($p < 0.001$). Avoidance coping demonstrated no significant association with functional scores ($p > 0.05$) but was positively

**Table 2. The EORTC QLQ-C30, PSS-10 and MCMQ scores of the participants.**

| Fields | Total Score | Items | Score |
|---|---|---|---|
| EORTC QLQ-C30 | | | |
| Functional fields | 64.81 ± 21.83 | Physical function | 71.12 ± 24.57 |
| | | Role function | 61.52 ± 33.19 |
| | | Emotional function | 67.04 ± 25.43 |
| | | Cognitive function | 75.75 ± 24.68 |
| | | Social function | 48.60 ± 31.09 |
| Symptomatic fields | 24.95 ± 19.38 | Fatigue | 40.01 ± 25.34 |
| | | Nausea and vomiting | 13.39 ± 20.05 |
| | | Pain | 21.44 ± 24.30 |
| Single-item fields | | Short breath | 20.60 ± 24.29 |
| | | Sleeplessness | 30.15 ± 32.03 |
| | | Appetite loss | 27.15 ± 28.22 |
| | | Constipation | 18.35 ± 25.80 |
| | | Diarrhoea | 9.18 ± 17.98 |
| | | Financial difficulty | 61.24 ± 35.46 |
| Overall life quality | 55.99 ± 24.64 | Overall health status | 55.99 ± 24.64 |
| PSS-10 | 19.63 ± 4.35 | Crisis perception | 11.06 ± 3.90 |
| MCMQ | | Coping ability | 8.57 ± 2.20 |
| | | confrontation | 18.87 ± 4.02 |
| | | Avoidance | 16.87 ± 2.69 |
| | | Yielding | 9.56 ± 3.27 |

**Table 3. Univariate analysis of the quality of life among patients with hematological cancer.**

| Patients' Characteristics | Number of cases | | Overall life quality | | | Functional fields | | | Symptomatic fields | | |
|---|---|---|---|---|---|---|---|---|---|---|---|
| | | | Score [$M(P_{25},P_{75})$] | Z/H | P | Score [$M(P_{25},P_{75})$] | Z/H | P | Score [$M(P_{25},P_{75})$] | Z/H | P |
| Age | <60 | 103 | 58.33 (50.00,83.33) | −2.337 | 0.019 | 70.33 (52.00,83.33) | −0.265 | 0.791 | 18.52 (11.11,33.33) | −0.606 | 0.554 |
| | 60~ | 75 | 50.00 (33.33,66.67) | | | 69.67 (50.67,80.33) | | | 22.22 (11.11,36.11) | | |
| Sex | Male | 106 | 58.33 (50.00,75.00) | −2.215 | 0.027 | 73.33 (58.67,85.33) | −3.228 | 0.001 | 16.67 (7.41,31.48) | −2.371 | 0.018 |
| | Female | 72 | 50.00 (33.33,66.67) | | | 63.00 (40.67,74.33) | | | 25.93 (12.96,35.19) | | |
| Marital status | Married | 151 | 50.00 (41.67,66.67) | 11.321 | 0.010 | 69.67 (50.67,82.67) | 2.802 | 0.423 | 20.37 (11.11,33.33) | 3.379 | 0.337 |
| | Unmarried | 11 | 66.67 (62.50,83.33) | | | 73.33 (67.50,76.33) | | | 18.52 (7.41,30.56) | | |
| | Divorced | 3 | 0(0,25.00) | | | 32.67 (16.33,54.67) | | | 57.41 (37.04,62.04) | | |
| | Widowed | 13 | 58.33 (50.00,66.67) | | | 71.00 (58.00,80.67) | | | 25.93 (22.22,31.48) | | |
| Place of Residence | City | 47 | 58.33 (50.00,66.67) | 9.834 | 0.020 | 71.33 (62.33,84.67) | 5.589 | 0.133 | 14.81 (10.19,29.63) | 7.556 | 0.056 |
| | County | 27 | 66.67 (50.00,79.17) | | | 72.33 (59.50,85.50) | | | 16.67 (10.19,26.85) | | |
| | Towns and villages | 10 | 54.17 (33.33,66.67) | | | 71.50 (54.67,84.00) | | | 18.52 (14.81,25.93) | | |
| | Village | 94 | 50.00 (33.33,66.67) | | | 66.33 (45.00,78.33) | | | 25.93 (12.96,40.74) | | |
| Type of insurance | Urban medical insurance | 51 | 66.67 (50.00,75.00) | 19.006 | 0.001 | 71.33 (64.17,84.67) | 10.96 | 0.004 | 16.67 (11.11,29.63) | 11.68 | 0.003 |
| | New rural cooperative medical system (NCMS) | 108 | 50.00 (33.33,66.67) | | | 63.33 (46.67,78.00) | | | 25.93 (13.89,38.89) | | |
| | Other | 19 | 75.00 (58.33,83.33) | | | 75.67 (69.00,88.50) | | | 14.81 (7.41,21.30) | | |

correlated with symptom burden($p<0.001$). The stress perception, crisis perception, and yielding exhibited a signifcant positive correlation with symptom scores ($p<0.001$). As shown in Table 4.

### 3.5 Multiple linear regression analysis of factors related to QoL in patients with hematological diseases

The total score of overall QoL, functional fields and symptomatic fields were used as the dependent variables, and the statistically significant variables in the univariate and correlation analyses were used as the independent variables in the multiple linear regression analysis ($\alpha_{in}$ = 0.05, $\alpha_{out}$ = 0.10). The independent variables were assigned as shown in Table 5.

The results of the multiple linear regression analysis revealed that sex, crisis perception and yielding were the main factors influencing the overall QoL and functional field of patients with hematological diseases ($p<0.05$), which explained 36.5% and 45.6% of the total variance in overall QoL. The results of the multiple linear regression analysis revealed that place of residence, crisis perception, avoidance and yielding were the main factors influencing the symptomatic fields of patients with hematological diseases ($p<0.05$), which explained 34.2% of the total variance in the QoL of symptomatic patients, however, there was no multicollinearity among the variables. As shown in Table 6. All variance inflation factors

**Table 4. Correlation analysis of quality of life, awareness of stress and medical coping strategies.**

| Items | Overall life quality | Functional fields | Symptomatic fields |
|---|---|---|---|
| PSS-10 | | | |
| PSS-10 total score | −0.389** | −0.591** | 0.435** |
| Crisis perception | −0.489** | −0.638** | 0.535** |
| Coping ability | 0.063 | −0.064 | −0.075 |
| MCMQ | | | |
| Confrontation | 0.121 | 0.149* | −0.028 |
| Avoidance | 0.039 | −0.068 | 0.280** |
| Yielding | −0.479** | −0.547** | 0.472** |

** $p < 0.001$; * $p < 0.05$

**Table 5. Independent variable assignment methods.**

| Independent variable | Assignment method |
|---|---|
| Gender | Male = 1, Female = 2 |
| Age (years) | <60 = 1, ≥60 = 2 |
| Marital status | Married = 1, Unmarried = 2, Divorced = 3, Widowed = 4 |
| Place of Residence | City = 1,County = 2,Towns and villages = 3,Village = 4 |
| Type of insurance | Urban medical insurance = 1,New rural cooperative medical system (NCMS)=2,Other = 3 |

**Table 6. Multivariate regression analysis of quality of life outcomes in patients with hematological malignancies.**

| Variables | Overall QoL | | | Symptomatic Fields | | | Functional Fields | | |
|---|---|---|---|---|---|---|---|---|---|
| | β (SE) | Std. β | p-value | β (SE) | Std. β | p-value | β (SE) | Std. β | p-value |
| Constant | —(17.047) | — | <0.001 | −0.082 (13.645) | — | 0.250 | (13.981) | — | <0.001 |
| Sex | 0.137 (3.215) | 0.137 | 0.034 | −0.082 (2.574) | −0.082 | 0.211 | 0.144 (2.637) | 0.144 | 0.016 |
| Age | −0.094 (3.211) | −0.094 | 0.146 | −0.025 (2.571) | −0.025 | 0.708 | 0.058 (2.634) | 0.058 | 0.329 |
| Marital status | 0.115 (1.972) | 0.115 | 0.074 | −0.026 (1.579) | −0.026 | 0.696 | 0.056 (1.618) | 0.056 | 0.351 |
| Place of Residence | −0.138 (1.344) | −0.138 | 0.055 | 0.151 (1.076) | 0.151 | 0.040 | −0.080 (1.103) | −0.080 | 0.232 |
| Type of insurance | 0.066 (2.869) | 0.066 | 0.351 | −0.083 (2.296) | −0.083 | 0.249 | 0.030 (2.353) | 0.030 | 0.639 |
| Crisis perception | −0.223 (0.494) | −0.223 | 0.005 | 0.218 (0.395) | 0.218 | 0.007 | −0.347 (0.405) | −0.347 | <0.001 |
| Coping ability | 0.091 (0.739) | 0.091 | 0.168 | −0.082 (0.592) | −0.082 | 0.227 | 0.016 (0.606) | 0.016 | 0.793 |
| Confrontation | 0.024 (0.399) | 0.024 | 0.710 | 0.003 (0.319) | 0.003 | 0.961 | 0.026 (0.327) | 0.026 | 0.670 |
| Avoidance | 0.045 (0.605) | 0.045 | 0.498 | 0.172 (0.484) | 0.172 | 0.011 | 0.011 (0.496) | 0.011 | 0.861 |
| Yielding | −0.348 (0.594) | −0.348 | <0.001 | 0.345 (0.476) | 0.345 | <0.001 | −0.362 (0.487) | −0.362 | <0.001 |

Overall QoL: $R^2 = 0.365$, Adj. $R^2 = 0.327$, F = 9.611, $p < 0.001$. Symptomatic Fields: $R^2 = 0.342$, Adj. $R^2 = 0.302$, F = 8.664, $p < 0.001$. Functional Fields: $R^2 = 0.456$, Adj. $R^2 = 0.423$, F = 13.975, $p < 0.001$. β (SE) = Unstandardized coefficient (Standard Error); Std. β = Standardized coefficient.

(VIFs) were <2.0, indicating no multicollinearity among predictors. For the 95% confidence intervals (CIs) of all variables in the models, please refer to the supplementary materials (see S2 Table).

## 4 Discussion

Our study demonstrates that patients with hematological malignancies experience significantly impaired QoL, as indicated by a mean overall score of 55.99±24.6, which falls within the moderate-to-low range compared to established

benchmarks in cancer populations. Functional domains were notably affected, particularly social functioning, while fatigue emerged as the most burdensome symptom. The strong negative correlations between perceived stress, crisis perception, yielding coping strategies and both QoL and functional scores demonstrate the profound impact of psychological factors on patient well-being. Notably, multivariate analysis revealed that sex, crisis perception, and yielding coping strategies significantly predicted overall QoL and functional status (explaining 36.5–45.6% of variance), while residence, crisis perception, and coping (particularly avoidance and yielding) emerged as primary determinants of symptom burden (accounting for 34.2% of variance). These findings emphasize the urgent need for integrated psychosocial interventions targeting stress management and adaptive coping strategies in clinical care to improve patient well-being.

QoL encompasses functional status (physical, role, cognitive, emotional, social), disease- and treatment-related symptoms, and global perceptions of health and well-being. In hematologic malignancies, QoL reflects not only physical symptoms but also treatment-related distress (e.g., anxiety about prognosis), social isolation due to immunosuppression, and financial toxicity from prolonged therapies [28]. The EORTC QLQ-C30 scale, widely used in oncology research, provides a comprehensive assessment of these dimensions, making it a valuable tool for evaluating QoL in hematological malignancies [22,29–31]. Our results are consistent with previous research indicating that hematological malignancies significantly impair QoL, particularly in social functioning and symptom burden [8]. The high fatigue scores observed in our study are also corroborated by earlier studies, emphasizing the pervasive nature of this symptom in cancer populations [32,33]. These comparisons highlight the universal challenges faced by hematological cancer patients and the need for standardized QoL assessments in clinical practice.

Sex differences in QoL were another critical finding, with female patients reporting lower functional scores and higher symptom burden compared to males. Compared with men, women's psychological characteristics may render their inner emotions more susceptible [34,35]. Our study showed that women have lower scores in functional areas and higher scores in symptom areas, leading to varying degrees of impact on the QoL of patients of different genders,which is consistent with the findings by Tinsley-Vance [36]. It is well established that the male sex is associated with increased risk for, as well as poorer survival of, most cancers. In terms of gender, the incidence and death of hematologic malignancies are generally higher in males than in females globally [2]. A recent study showed that they identified 36,795 lymphoma cases, 20,738 (56.4%) in men and 16,057 (43.6%) in women [37]. Our data showed that 40.4% of the participants were female is similar to this trend. These disparities may stem from biological, psychological, or sociocultural factors, warranting further investigation to tailor gender-specific supportive care.

Perceived stress refers to an individual's cognitive and emotional responses to the stressors they face. Previous studies has shown that perceived stress in patients with hematological malignancies is negatively correlated with QoL, meaning that higher levels of perceived stress are associated with lower QoL [10]. High perceived stress is recognized as a major life stressor in cancer [38], capable of triggering psychological disorders (e.g., anxiety, depression) and potentially influencing disease progression via immune pathways, ultimately impacting QoL and treatment adherence. Our results confirm its significant negative impact on both functional and symptom domains in hematological malignancies. Therefore, it is crucial for healthcare providers to recognize and address these psychological needs through comprehensive support and evidence-based interventions, such as cognitive behavioral therapy (CBT), mindfulness-based approaches, relaxation techniques, and group therapy [39,40], to improve overall well-being. Future research should focus on developing and implementing effective interventions to manage perceived stress and its associated symptoms, thereby enhancing the QoL of patients with hematological malignancies.

This study showed that the coping strategies employed by patients with hematological malignancies also impact their QoL. Avoidant coping can adversely affect survivors' ability to focus on solutions and necessary actions. While avoidance coping showed no direct benefit for functional status, its positive association with symptom burden. Patients who employ avoidant coping strategies show less pronounced symptoms, which may be related to the fact that avoidance can, to some extent, alleviate stress responses [41]. However, chronic avoidance impedes symptom reporting and treatment

adherence, ultimately exacerbating physical distress as seen in our symptom models. Our findings are consistent with other reports. Previous studies have shown that avoidant coping was associated with distress, and significantly mediated the relationship between each well-being variable (fear of recurrence, attention, body image, fatigue, social support, and social constraints) and each distress indicator (depression and anxiety) [42]. Specifically, avoidant coping has been associated with depression, anxiety, cancer-related worries, and generalized distress [43].

The yielding exhibited a negative correlation with QoL function scores. The crisis perception and yielding exhibited a significant positive correlation with symptom scores. Patients who yield to illness often reduce their compliance with disease treatment, delay treatment, adopt a negative attitude towards life, passively accept treatment without purpose, leading to economic toxicity and even treatment interruption, which in turn leads to disease progression and affects the quality of life [44]. Yielding is a negative coping strategy which is negatively correlated with overall health and functional domains and positively correlated with symptom domains. The more patients submit to their illness, the more pronounced their symptoms become, the lower their treatment motivation, the poorer their functional status, and the lower their QoL [45]. Therefore, healthcare professionals should strengthen communication with patients, gain a deep understanding of their symptom burden and coping strategies, and provide personalized disease management knowledge, which can help improve patients' compliance.

## 5 Limitation

While the present study provides valuable insights into the relationship between QoL, perceived stress, and medical coping in patients with hematological malignancies, it is not without its limitations. These limitations should be considered when interpreting the results and conclusions. The relatively small sample size from a single center limits the generalizability of the findings to the broader population of patients with hematological malignancies. Future studies should include larger, multi-center samples with greater diversity (e.g., in socioeconomic status, ethnicity, and geographic region) to enhance generalizability. The study employed a cross-sectional design, which limits the ability to establish causal relationships between psychological state, coping strategies, and QoL. Longitudinal studies are needed to better understand the temporal dynamics and causal pathways. Additionally, the cross-sectional design precludes assessment of changes in QoL over time. Future longitudinal research is warranted to track QoL trajectories and evaluate the effectiveness of personalized interventions tailored to patients' specific profiles for improving QoL and prognosis.

## 6 Conclusions

In summary, this study revealed that QoL in patients with hematological malignancies is generally suboptimal and significantly associated with sex, crisis perception, and maladaptive coping strategies (particularly yielding). Targeted interventions focusing on these modifiable factors, especially enhancing stress management and adaptive coping skills, hold promise for improving QoL. Specifically, oncology nurses play a crucial role in providing tailored health education, psychological support, and addressing patients' psychosocial needs to empower them in managing their illness and enhancing well-being. This study contributes to the growing evidence on psychosocial determinants of QoL in oncology and underscores the importance of integrating tailored supportive care programs into the management of hematological malignancies.

## Supporting information

**S1 Table. STROBE statement—checklist of items that should be included in reports of observational studies.**
(DOCX)

**S2 Table. The 95% confidence intervals (CIs) of all variables in the models.** Note: Overall QoL: $R^2 = 0.365$, Adj. $R^2 = 0.327$, $F = 9.611$, $p < 0.001$. Functional Fields: $R^2 = 0.456$, Adj. $R^2 = 0.423$, $F = 13.975$, $p < 0.001$. Symptomatic Fields: $R^2 = 0.342$, Adj. $R^2 = 0.302$, $F = 8.664$, $p < 0.001$.
(DOCX)

**S1 Data. The dataset used for analysis.**
(XLSX)

## Acknowledgments

We are thankful for the generous contributions of the research participants and the staff who assisted with data collection during the study.

## Author contributions

**Data curation:** Yue Qi, Wei Liu.

**Formal analysis:** Yue Qi, Kangsheng Zhu.

**Funding acquisition:** Yue Qi.

**Investigation:** Wei Liu.

**Methodology:** Kangsheng Zhu.

**Project administration:** Xiaolei Xu.

**Software:** Yue Qi.

**Supervision:** Wei Liu.

**Validation:** Yue Qi, Xiaolei Xu.

**Writing – original draft:** Yue Qi, Kangsheng Zhu, Xiaolei Xu.

**Writing – review & editing:** Yue Qi, Kangsheng Zhu, Xiaolei Xu.

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
