## [Decision Letter · Decision Letter 0]

28 May 2025

PONE-D-25-00806Relationships between perceived stress, coping strategies and quality of life in patients with hematological malignancies: A cross-sectional studyPLOS ONE

Dear Dr. Xu

Thank you for submitting your manuscript to PLOS ONE. After careful consideration, we feel that it has merit but does not fully meet PLOS ONE’s publication criteria as it currently stands. Therefore, we invite you to submit a revised version of the manuscript that addresses the points raised during the review process.

We look forward to receiving your revised manuscript.

Kind regards,

Othman A. Alfuqaha, Ph.D.

Academic Editor

PLOS ONE

Journal Requirements:

“This paper was partially funded by an unconditional grant from the Medical Science Research Project Plan for 2024 (Project number 20241949).”

“Medical Science Research Project of hebei for 2024 (Project number 20241949)”

Additional Editor Comments (if provided):

Dear Authors,

Thank you for submitting your manuscript to our journal. After careful review, we have some major comments that need to be addressed before we can consider your paper for publication. Please find the comments below:

Abstract

1. Method Section: Please add details about the statistical analysis tests used in your study.

2. Results: Provide context for the mean score of 55.99, indicating whether it represents a high, moderate, or low level of the measured construct. Also, include the specific numbers for the negative correlation.

Conclusion

1. Broad Implications: Please provide a broader implication for your study, highlighting its potential applications and contributions.

Introduction

1. Importance of the Study: Please clarify why your study is important, despite previous studies addressing similar issues. Highlight the unique contributions and gaps your study addresses.

2. Additional References: Enrich the introduction with more references to provide a comprehensive background and context for your research question.

3. Study Question: Consider adding a clear study question to guide the reader through the purpose and objectives of your research.

Methods

1. STROBE Guidelines: Please follow the STROBE guidelines for reporting observational studies, ensuring that your methodology is transparent and comprehensive.

2. Sample Size Calculation: Provide details on how the sample size was calculated to ensure it was sufficient for your analysis. Clarify the response rate and how you handled missing data.

3. Generalizability: Discuss the limitations of generalizing results from a single hospital and provide justification for your study setting and population.

4. Consent Form and Study Design: Clarify the consent process, study design, and sampling procedure, addressing the "Why, What, How, Where, When" aspects of your methodology.

5. Study Tool Details: Add subheadings for the study tool section, specify the language used, and include Cronbach's alpha values to demonstrate the reliability of your instruments.

6. Statistical Analysis: Provide more detailed explanations of the statistical analysis to make it accessible to readers.

Results

1. Section Formatting: Replace abbreviations with words for better readability. Consider adding a column to interpret the score meanings (high, moderate, low) based on cut-off criteria.

2. Tables: Consider merging tables to improve readability and flow. Add p-values as footnotes where appropriate.

Discussion

1. Highlighting Results: Highlight your key findings and their significance in the first paragraph of the discussion.

2. Definition of Quality of Life (QoL): Provide a comprehensive definition of QoL, aligning with established definitions in the literature.

3. Referencing: Add references to compare your results with previous studies, enhancing the discussion and contextualizing your findings.

Conclusion Section

Please add a conclusion section that summarizes the main findings and implications of your study.

References

Double-check and format the references according to our journal's guidelines.

Editing and Proofreading

Please have your manuscript edited and proofread by a native English speaker to ensure clarity, coherence, and adherence to professional standards.

We look forward to receiving your revised manuscript.

Best regards,

Reviewers' comments:

Reviewer's Responses to Questions

**Comments to the Author**

1. Is the manuscript technically sound, and do the data support the conclusions?

Reviewer #1: Partly

2. Has the statistical analysis been performed appropriately and rigorously? 

Reviewer #1: N/A

3. Have the authors made all data underlying the findings in their manuscript fully available?

Reviewer #1: Yes

4. Is the manuscript presented in an intelligible fashion and written in standard English?

Reviewer #1: No

5. Review Comments to the Author

Reviewer #1: General Comments:

The study examines the relationship between perceived stress, coping strategies, and quality of life in patients with hematological malignancies. The topic is relevant and important for clinical and psychological research. However, there are several areas for improvement in clarity, methodological justification, coherence, and language precision.

Specific Comments:

Title: The title could be more precise: “Relationships Between Perceived Stress, Coping Strategies, and Quality of Life in Patients with Hematological Malignancies: A Cross-Sectional Study” is slightly lengthy. Consider revising it to be more concise, e.g., "Impact of Stress and Coping Strategies on Quality of Life in Hematological Malignancies".

Abstract:

Line 4: “It is not clear how stress perceptions, and medical coping strategies affect the quality of life (QoL) with hematological malignancies.” The phrase "stress perceptions" is awkward; consider "perceived stress."

Can you rewrite like this? “The impact of perceived stress and medical coping strategies on the quality of life (QoL) in patients with hematological malignancies is not well understood.”

Line 9: "Additionally, we explored the associations between stress perceptions, medical coping strategies and QoL." Missing comma before "and."

Line 16: "The avoidance exhibited a significant positive correlation with unction scores (p<0.05)." Typo in “unction” (should be "function").

Introduction:

Paragraph 1:

Line 2: "These diseases account for approximately 9% of all cancers, ranking as the fourth most common type of malignancy." Needs a citation.

Line 7: “These factors will increase the patients' economic burden and cause severe psychological disorders.” The phrase "severe psychological disorders" may be too strong without evidence. Consider "psychological distress."

Paragraph 2:

Line 3: "Studies have shown a correlation between perceived stress and QoL, indicating that the more negative perceptions of the disease, the poorer the QoL." Needs references. Also, rephrase for clarity: “Studies indicate that higher perceived stress is associated with lower QoL.”

Methods:

Study Design & Population

Line 3: "The study, involving 185 hematologic cancer patients in China, was conducted between 8, 2024 to 12, 2024." Incorrect date format. Should be “from August 2024 to December 2024.”

Line 10: “Patients who were fully cognizant of their own medical condition.” "Fully cognizant" is vague. Does this mean cognitively intact? Consider “Patients who were aware of their diagnosis and treatment.”

Questionnaire survey and ethical considerations

Line 7: "We did not involve minors and all them received written informed consent." Grammar error, should be “We did not include minors, and all participants provided written informed consent.”

Statistical Analysis:

Line 5: "p<0.05 was considered statistically significant." The manuscript does not mention multiple testing correction (e.g., Bonferroni adjustment). Were adjustments made?

Results:

Table 1 & Demographics

The manuscript states that 40.4% of the participants were female but does not compare this distribution to the general population of hematological malignancies. Was gender distribution representative?

Correlation Analysis (Table 4):

Line 6: “The avoidance exhibited a significant positive correlation with function scores (p<0.05).” The interpretation is unclear—avoidance typically has a negative impact on QoL. More discussion is needed.

Regression Analysis (Tables 6–8):

Line 9: "Crisis perception and yielding were the main factors influencing overall QoL." The manuscript should explore why these factors are significant—what mechanisms are at play?

Discussion:

Interpretation of Results:

Paragraph 2: The section discussing gender differences in QoL is interesting but lacks references to studies supporting these findings.

Paragraph 4: “Therefore, it is crucial for healthcare providers to recognize and address the psychological needs of these patients.” This conclusion is important but should be more specific—what types of psychological interventions are recommended?

Limitations:

Line 4: “Future studies should aim to include a larger and more diverse sample.” Good point, but clarify what aspects of diversity (e.g., socioeconomic status, ethnicity, or geographic region) are lacking in this study.

Conclusion:

Line 3: "Strengthening the attention in this factors will help to improve the QoL." Incorrect grammar. Should be “Strengthening attention to these factors may help improve QoL.”

6. PLOS authors have the option to publish the peer review history of their article (what does this mean? ). If published, this will include your full peer review and any attached files.

**Do you want your identity to be public for this peer review?** For information about this choice, including consent withdrawal, please see our Privacy Policy .

Reviewer #1: **Yes: ** Sunil Shrestha

---

## [Author Response · Author response to Decision Letter 1]

12 Jul 2025

Dear Editor and Reviewer,

We are very grateful for the time and effort in evaluating our manuscript “Relationships Between perceived stress, coping strategies and Quality of Life in patients with hematological malignancies: A Cross-Sectional Study”. In response to the academic editor and reviewers' comments and suggestions, we have thoroughly revised our paper. Below we included a detailed point-by-point response to the comments. The changes we have made to the manuscript are highlighted in yellow.

Response to the academic editor:

Response: We sincerely appreciate your valuable comments. We have revised the manuscript in accordance with PLOS ONE's style requirements

Response:

Our research is a cross-sectional survey, and the article contains all the original data needed to replicate the research results. Its form is presented in terms of mean, standard deviation, 95% confidence interval, etc. Furthermore, we uploaded the original data as an attachment in the Excel table.

Response:

We have an ORCID ID (0000-0002-4999-1533) and it has been validated in Editorial Manager. The ORCID site: https://orcid.org/0000-0002-4999-1533

4. Thank you for stating the following in the Acknowledgments Section of your manuscript:“This paper was partially funded by an unconditional grant from the Medical Science Research Project Plan for 2024 (Project number 20241949).”We note that you have provided additional information within the Acknowledgements Section that is not currently declared in your Funding Statement. Please note that funding information should not appear in the Acknowledgments section or other areas of your manuscript. We will only publish funding information present in the Funding Statement section of the online submission form.

“Medical Science Research Project of hebei for 2024 (Project number 20241949)”

Response: Thank you very much for your comments.

We have removed the funding-related text from the manuscript in the Acknowledgements Section.

Our Funding Statement reads as follows:

This work was supported by the Medical Science Research Project of Hebei (grant no. 20241949).

Response to Additional Editor Comments:

Abstract

1. Method Section: Please add details about the statistical analysis tests used in your study.

2. Results: Provide context for the mean score of 55.99, indicating whether it represents a high, moderate, or low level of the measured construct. Also, include the specific numbers for the negative correlation.

Conclusion

1. Broad Implications: Please provide a broader implication for your study, highlighting its potential applications and contributions.

Response: Thank you for your constructive comments. We have carefully addressed the points in the revised version. The modifications include: Added detailed statistical methods in the Abstract's Method section. In the Results section of the Abstract, we clarified that the mean score of 55.99 indicating a moderate-to-low level compared to established benchmarks in cancer populations. We added the specific values for the correlation. ( see lines 27-28,31-39 )

-Methods

1. STROBE Guidelines: Please follow the STROBE guidelines for reporting observational studies, ensuring that your methodology is transparent and comprehensive.

2. Sample Size Calculation: Provide details on how the sample size was calculated to ensure it was sufficient for your analysis. Clarify the response rate and how you handled missing data.

3. Generalizability: Discuss the limitations of generalizing results from a single hospital and provide justification for your study setting and population.

4. Consent Form and Study Design: Clarify the consent process, study design, and sampling procedure, addressing the "Why, What, How, Where, When" aspects of your methodology.

5. Study Tool Details: Add subheadings for the study tool section, specify the language used, and include Cronbach's alpha values to demonstrate the reliability of your instruments.

6. Statistical Analysis: Provide more detailed explanations of the statistical analysis to make it accessible to readers.

Response: Thank you for your constructive comments.

1 We appreciate your suggestion to improve the reporting quality of our observational study. As recommended, we have carefully revised our manuscript to fully comply with the STROBE guidelines. In addition, we included the completed STROBE Checklist as a supplementary file to facilitate verification of our adherence to reporting standards. ( see 2.1 Study design and ethical considerations)

2 Thank you for your comment. We confirm that the sample size calculation has already been described in detail in the Methods section of our manuscript. Specifically, we stated: The sample size for this multivariate linear regression analysis was determined based on the standard criterion requiring 10-20 participants per independent variable. This study included 10 primary predictors; thus, the estimated sample size ranged between 100 and 200 participants. To account for a potential 10% loss to follow-up or incomplete responses, we distributed 185 questionnaires, ensuring adequate statistical power for model stability. ( see lines 180-184 )

3. Although a single-center sample may limit the wide applicability of the results, our hospital is one of the larger hematological oncology specialties in northern China. It is a member of the Lymphoma Collaboration Group of Northern China and the Lymphoma Working Group of China, which has a substantial pool of hematology patients and follows standardized treatment protocols, while strictly controlling for confounding variables to enhance the reliability of causal inferences. ( see lines 98-104 )

4 We added the consent process, study design and sampling procedure in the method section.

5 We specified the language used and included Cronbach's alpha values. ( see 2.4 Measurements)

Results

1. Section Formatting: Replace abbreviations with words for better readability. Consider adding a column to interpret the score meanings (high, moderate, low) based on cut-off criteria.

2. Tables: Consider merging tables to improve readability and flow. Add p-values as footnotes where appropriate.

Response: Thank you for your comment. We have replaced some abbreviations with words in results for better readability. We merged Tables 6/7/8. Due to the limitations of the tables, other data have been presented as supplementary information. (see Results 3.5 and Table 6)

Discussion

1. Highlighting Results: Highlight your key findings and their significance in the first paragraph of the discussion.

2. Definition of Quality of Life (QoL): Provide a comprehensive definition of QoL, aligning with established definitions in the literature.

3. Referencing: Add references to compare your results with previous studies, enhancing the discussion and contextualizing your findings.

Response: We have revised the Discussion section to emphasize our key findings and their significance in the opening paragraph, as suggested. As requested, we have expanded the second paragraph to include a comprehensive definition of quality of life, supported by citations from established literature. Additionally, we have incorporated comparative analyses with other similar studies to contextualize our findings and strengthen the discussion.(see Discussion)

Conclusion Section

Please add a conclusion section that summarizes the main findings and implications of your study.

References

Double-check and format the references according to our journal's guidelines.

Editing and Proofreading

Please have your manuscript edited and proofread by a native English speaker to ensure clarity, coherence, and adherence to professional standards.

Response:

As suggested, we have strengthened the conclusion by highlighting the study's main outcomes and providing more detailed interpretation of their importance.

All the revised sentences are highlighted in yellow.

Response to Reviewer #1

Title: The title could be more precise: “Relationships Between Perceived Stress, Coping Strategies, and Quality of Life in Patients with Hematological Malignancies: A Cross-Sectional Study” is slightly lengthy. Consider revising it to be more concise, e.g., "Impact of Stress and Coping Strategies on Quality of Life in Hematological Malignancies".

Response: We sincerely thank you for identifying the need for a more precise title. After careful consideration, we have adopted your suggested approach and present the improved title: Impact of Stress and Coping Strategies on Quality of Life in Hematological Malignancies : A cross-sectional study. As this is a questionnaire-based study, we have revised the title to clearly reflect the research methodology while maintaining conciseness. (see the title)

Abstract:

Line 4: “It is not clear how stress perceptions, and medical coping strategies affect the quality of life (QoL) with hematological malignancies.” The phrase "stress perceptions" is awkward; consider "perceived stress."

Can you rewrite like this? “The impact of perceived stress and medical coping strategies on the quality of life (QoL) in patients with hematological malignancies is not well understood.”

Line 9: "Additionally, we explored the associations between stress perceptions, medical coping strategies and QoL." Missing comma before "and."

Line 16: "The avoidance exhibited a significant positive correlation with unction scores (p<0.05)." Typo in “unction” (should be "function").

Response: Thank you for your suggestion. We agree that "perceived stress" is more precise terminology. We have systematically replaced all instances of 'stress perceptions' with 'perceived stress' throughout the manuscript to maintain terminological consistency. We have proofread the manuscript to correct punctuation errors and spelling mistakes throughout the text.

Introduction:

Paragraph 1:

Line 2: "These diseases account for approximately 9% of all cancers, ranking as the fourth most common type of malignancy." Needs a citation.

Line 7: “These factors will increase the patients' economic burden and cause severe psychological disorders.” The phrase "severe psychological disorders" may be too strong without evidence. Consider "psychological distress."

Paragraph 2:

Line 3: "Studies have shown a correlation between perceived stress and QoL, indicating that the more negative perceptions of the disease, the poorer the QoL." Needs references. Also, rephrase for clarity: “Studies indicate that higher perceived stress is associated with lower QoL.”

Response: Thank you for your suggestion. After our careful thinking, we rewrote this sentence and cited the corresponding literature. We fully accept your suggestion. "severe psychological disorders" is indeed somewhat inappropriate and we have changed it to "psychological distress". (see line 69)

Methods:

Study Design & Population

Line 3: "The study, involving 185 hematologic cancer patients in China, was conducted between 8, 2024 to 12, 2024." Incorrect date format. Should be “from August 2024 to December 2024.”

Line 10: “Patients who were fully cognizant of their own medical condition.” "Fully cognizant" is vague. Does this mean cognitively intact? Consider “Patients who were aware of their diagnosis and treatment.”

Questionnaire survey and ethical considerations

Line 7: "We did not involve minors and all them received written informed consent." Grammar error, should be “We did not include minors, and all participants provided written informed consent.”

Response: Thank you for your suggestion. We have corrected the date format: from August 2024 to December 2024. Thank you for your careful review. We fully accept your suggestion and have changed these phrases to "Patients who were aware of their diagnosis and treatment" and "We did not include minors, and all participants provided written informed consent." (see lines 96-97,116,106-107)

Statistical Analysis:

Line 5: "p<0.05 was considered statistically significant." The manuscript does not mention multiple testing correction (e.g., Bonferroni adjustment). Were adjustments made?

Results:

Table 1 & Demographics

The manuscript states that 40.4% of the participants were female but does not compare this distribution to the general population of hematological malignancies. Was gender distribution representative?

Correlation Analysis (Table 4):

Line 6: “The avoidance exhibited a significant positive correlation with function scores (p<0.05).” The interpretation is unclear—avoidance typically has a negative impact on QoL. More discussion is needed.

Regression Analysis (Tables 6–8):

Line 9: "Crisis perception and yielding were the main factors influencing overall QoL." The manuscript should explore why these factors are significant—what mechanisms are at play?

Response: Thank you for your comment. In this study, Post hoc pairwise comparisons were conducted using Bonferroni adjustment for multiple comparisons.

It is well established that the male sex is associated with increased risk for, as well as poorer survival of, most cancers. In terms of gender, the incidence and death of hematologic malignancies are generally higher in males than in females globally (Blood Cancer J. 2023 May 17;13(1):82.). A recent study (Am J Hematol. 2023 Jan;98(1):23-30.) showed that they identified 36,795 lymphoma cases, 20,738 (56.4%) in men and 16,057 (43.6%) in women. Our data showed that 40.4% of the participants were female is similar to this trend. We elaborated in the discussion section.(see lines 198-199, 316-320)

We sincerely appreciate the reviewer's astute observation regarding the interpretation of avoidance coping. We acknowledge that the original statement ("The avoidance exhibited a significant positive correlation with function scores") was imprecise and have revised this section for clarity. The complete corrections include: Our results actually showed: There was no significant correlation between avoidance and functional scores (r = -0.068, p=0.381 in Table 4). The original Line 6 has been modified to Avoidance coping demonstrated no significant association with functional scores (p>0.05) but was positively correlated with symptom burden(p<0.001). Meanwhile, we cited relevant literature and conducted in-depth discussions. For details, please refer to the discussion section. We deeply appreciate your insightful suggestion to elucidate the mechanisms underlying crisis perception and yielding coping. We have substantially expanded the Discussion section.(see lines 245-247,3287-334,359-362,)

Discussion:

Interpretation of Results:

Paragraph 2: The section discussing gender differences in QoL is interesting but lacks references to studies supporting these findings.

Paragraph 4: “Therefore, it is crucial for healthcare providers to recognize and address the psychological needs of these patients.” This conclusion is important but should be more specific—what types of psychological interventions are recommended?

Response: We deeply appreciate your insightful suggestion. We expanded the section discussing gender differences in QoL and cited relevant references, and recommended some psychological intervention measures, such as cognitive behavi

---

## [Editor Report · Decision Letter 1]

22 Aug 2025

Impact of Stress and Coping Strategies on Quality of Life in Hematological Malignancies : A cross-sectional study

PONE-D-25-00806R1

Dear Dr.

<table border="0" cellpadding="0" cellspacing="0" class="datatable3" style="border-collapse: collapse; width: 678px; line-height: 14px; color: rgb(0, 0, 51); font-family: verdana, geneva, arial, helvetica, sans-serif; font-size: 11.2px;"> <tbody> <tr style="background-color: rgb(244, 244, 244);"> <td style="padding: 3px; border: 1px solid rgb(255, 255, 255);">Xiaolei Xu</td> </tr> <tr style="background-color: rgb(244, 244, 244);"> <td style="padding: 3px; border: 1px solid rgb(255, 255, 255); width: 196.094px;"> </td> </tr> </tbody></table>

We’re pleased to inform you that your manuscript has been judged scientifically suitable for publication and will be formally accepted for publication once it meets all outstanding technical requirements.

Kind regards,

Othman A. Alfuqaha, Ph.D.

Academic Editor

PLOS ONE

Additional Editor Comments (optional):

As the Academic Editor, I have carefully reviewed the revised manuscript, and I am satisfied that all of my comments, as well as those raised by the reviewers, have been properly addressed.
---

## [Editor Report · Acceptance letter]

PONE-D-25-00806R1

PLOS ONE

Dear Dr. Xu,

I'm pleased to inform you that your manuscript has been deemed suitable for publication in PLOS ONE. Congratulations! Your manuscript is now being handed over to our production team.

Kind regards,

on behalf of

Dr. Othman A. Alfuqaha

Academic Editor

PLOS ONE